# Microbiological and Physicochemical Composition of Various Types of Homemade Kombucha Beverages Using Alternative Kinds of Sugars

**DOI:** 10.3390/foods11101523

**Published:** 2022-05-23

**Authors:** Maciej Ireneusz Kluz, Karol Pietrzyk, Miłosz Pastuszczak, Miroslava Kacaniova, Agnieszka Kita, Ireneusz Kapusta, Grzegorz Zaguła, Edyta Zagrobelna, Katarzyna Struś, Katarzyna Marciniak-Lukasiak, Jadwiga Stanek-Tarkowska, Adrian Vasile Timar, Czesław Puchalski

**Affiliations:** 1Department of Bioenergetics, Food Analysis and Microbiology, Institute of Food Technology and Nutrition, University of Rzeszow, Zelwerowicza 4, 35-601 Rzeszów, Poland; karolp@dokt.ur.edu.pl (K.P.); mkacaniova@ur.edu.pl (M.K.); akita@ur.edu.pl (A.K.); gzagula@ur.edu.pl (G.Z.); ezagrobelna@ur.edu.pl (E.Z.); kstrus@ur.edu.pl (K.S.); cpuchal@ur.edu.pl (C.P.); 2Department of Soil Science, Environmental Chemistry and Hydrology, University of Rzeszow, Zelwerowicza 8B, 35-601 Rzeszów, Poland; miloszp@dokt.ur.edu.pl (M.P.); jstanek@ur.edu.pl (J.S.-T.); 3Department of Food Technology and Human Nutrition, Institute of Food Technology and Nutrition, University of Rzeszow, Zelwerowicza 4, 35-601 Rzeszów, Poland; ikapusta@ur.edu.pl; 4Faculty of Food Assessment and Technology, Institute of Food Sciences, Warsaw University of Life Sciences (WULS-SGGW), Nowoursynowska st.159c, 02-776 Warsaw, Poland; katarzyna_marciniak_lukasiak@sggw.edu.pl; 5Food Engineering Department, University of Oradea, General Magheru 26, 410048 Oradea, Romania; atimar@uoradea.ro

**Keywords:** kombucha, MALDI-TOF MS, UPLC, ICP-OS, antibacterial properties, antioxidant properties

## Abstract

Kombucha is a beverage made by fermenting sweetened tea with a symbiotic culture of yeast and bacteria. Literature data indicate that the kombucha beverage shows many health-promoting properties such as detoxification, chemo-preventive, antioxidant, antimicrobial, antifungal, and general strengthening. The research conducted focuses on the analysis of polyphenolic compounds formed in the fermentation process using ultra-efficient liquid chromatography, as well as on checking the antimicrobial properties of kombucha against pathogenic bacteria and yeasts found in food. Analysis of the composition of the tea mushroom (SCOBY) microflora using the MALDI TOF MS Biotyper mass spectrometer showed 8 species of bacteria and 7 species of yeasts. In vitro studies confirm the bactericidal and bacteriostatic properties of fermented kombucha beverages, with white and green tea beverages showing the highest antibacterial activity. The bacteria *Staphylococcus aureus* and yeast *Candida albicans* were the most sensitive to the effects of kombucha tea beverages. UPLC chromatographic analysis confirmed the presence of 17 bioactive compounds in kombucha beverages that can affect human health. The analyses conducted were aimed at indicating the best recipe and conditions to prepare a kombucha beverage, which allowed the selection of the version with the best health-promoting properties. Fermented kombucha teas contain many elements such as aluminium, calcium, iron, potassium, magnesium, sodium, phosphorus, and sulphur.

## 1. Introduction

The kombucha mushroom is a symbiotic culture of acetic acid bacteria and yeast [1]. The tea mushroom produces a fermented tea beverage consisting of symbiotic cultures of bacteria and yeasts (SCOBY), sugars, and tea leaves, which make up a sour-flavoured carbonated beverage that is fermented for 7 to 30 days [1,2].

The kombucha beverages are characterized by a very rich composition. It contains, among others, organic acids (acetic, lactic, malic, ascorbic, gluconic, glucuronic, and oxalic), B vitamins (B1, B2, B6, and B12), vitamin C, minerals (zinc, magnesium, calcium, iron, and copper), ethanol (up to 2%), caffeine, carbon dioxide, proteins, fats, polyphenols, and sugars. The current strengths are the promising results that were obtained in vitro proving antioxidant, immunemodulatory, antiproliferative, hypocholesterolemic, antihypertensive, hypoglycemic, antimicrobial, etc., abilities. However, a significantly lower number of in vivo studies was carried out to confirm these effects [3].

The availability of kombucha tea in many forms and flavours, as well as the confirmation of its properties by numerous scientific research results, have contributed to its popularity and increasing availability [4].

The kombucha microbiome includes several types of acetic acid bacteria, yeast, and lactic acid bacteria [5]. Previous studies have shown bacteria of the genus *Acetobacter*, *Gluconobacter*, and *Komagataeibacter*, yeasts of the genus *Zygosaccharomyces* and *Brettanomyces* [6] known as symbiotic cultures of bacteria and yeast, so called SCOBYs. During metabolic reactions involving acetic acid bacteria in kombucha, sugars and alcohols are converted into, for example, acetic, succinic, gallic, and malic acid. Cellulose bacteria of the genera *Gluconoacetobacter* and *Komagataeibacter* are involved in the production of gluconic and glucuronic acid. Mutual metabolic interactions between bacteria and yeast species in kombucha affect the biological, physicochemical, and organoleptic properties of this beverage [7]. Previous research has shown that kombucha has antibacterial and antioxidant properties that positively affect the functioning of the immune system [6]. Furthermore, it has been shown to positively influence the regulation of sugar metabolism, demonstrating antidiabetic and detoxifying effects [6,8,9]. The detoxification properties of black tea kombucha of the body, resulting from the presence of glucuronic acid and its derivatives, show the ability to combine with toxins and drug metabolites, and these are then neutralized and excreted from the body, an antioxidant effect due to the content of vitamin C, polyphenols, and some minerals, supporting the immune system and influence on the reconstruction of the bacterial flora (thanks to the presence of lactic, acetic, and malic acids, vitamins, and minerals), regulating the digestive system, improving the work of the kidneys and bladder, and supporting the regeneration of the liver [10].

In addition, the kombucha beverage can be helpful for many ailments, e.g., insomnia, lack of appetite, memory loss, arterial hypertension, high cholesterol, obesity, weakness, chronic fatigue syndrome, premenstrual syndrome, rheumatism, arthritis, diabetes, and cancer [11,12].

The main objective of this work was to determinate of the chemical composition (bioactive compounds) and the microbiological SCOBY composition, mineral content, organoleptic analysis of new kinds of homemade kombucha beverages using alternative kinds of sugars (cane and coconut sugars).

## 2. Materials and Methods

### 2.1. Preparation of the Kombucha Beverages

Kombucha starter culture (inoculum) were obtained from our own collection from the Department of Bioenergetics, Food analysis, and Microbiology at the University of Rzeszow. The starter culture used in the study was stored in a refrigerator (4 °C ± 1 °C) and consisted of a sour broth and cellulosic layer (SCOBY floating on the liquid surface). Kombucha cultures were kept under aseptic conditions. The following teas (*Camellia sinensis*), were used in the work: black, green, and white teas from Kenia, Tanzania, and China, respectively, and carbon sources (sugars), i.e., cane and coconut bought at local supermarket. The teas were prepared in glass flasks. An infusion of 10 g per 1 L of water was made. The leaves were poured with hot water and infused for 10 min. Subsequently, various sugar sources were added at a concentration of 70 g/L. The sweetened infusion was allowed to cool to 22 °C and, after cooling, each sample was inoculated with 50 mL of kombucha from the previous fermentation and 10 g of SCOBY culture (Table 1). The fermentation process was conducted for 14 days in a dark room at 22 °C.

### 2.2. Antibacterial Activity of the Kombucha Beverages

To determine the bactericidal and bacteriostatic properties of kombucha beverages, an antibiogram was performed using the disc diffusion method. Eight strains of microorganisms were used in the experiment, including *Staphylococcus aureus* PCM 502, *Salmonella* sp. PCM 2548, *Escherichia coli* PCM 2209, *Listeria monocytogenes* PCM 2606, *Candida glabrata* PCM 2703-FY, *Candida albicans* PCM 1407, *Candida krusei* PCM 2706-FY, and *Candida tropicalis* PCM 2703-FY. All strains used for the investigation came from the Polish Collection of Microorganisms. In the first stage of the study, 0.1 mL of the test microorganism suspension with a density of 10^5^ CFU/ mL was applied to nutrient broth agar (Biocorp, Warsaw, Poland); 6 mm diameter filter paper discs were applied with 15 µL of tea extract and placed on inoculated agars. The Petri dishes were incubated aerobically for 24 h at 30 °C (for yeasts) and 37 °C (for bacteria). The diameters of the inhibition zones were measured in mm after incubation.

### 2.3. Identification of Kombucha Microflora Using by MALDI-TOF MS Biotyper

The 6 kombucha samples were selected for further testing after storage at 4 °C. The sample for the MALDI-TOF MS Biotyper analysis was prepared according to the extraction procedure provided by the manufacturer (Bruker Daltonik, Bremen, Germany). The bacterial and yeasts colony from SCOBYs from different kinds of kombucha were suspended in 300 µL of water (Sigma-Aldrich, St. Louis, MO, USA) and 900 µL of absolute ethanol (Bruker Daltonik, Bremen, Germany), mixed ten times, and centrifuged at 13,000 rpm for 2 min. The supernatant was rejected, and the pellets were centrifuged several times. After removal of the supernatant, the pellets were mixed with 10 µL of 70% formic acid (*v*/*v*) (Sigma-Aldrich, Saint Louis, MO, USA) and the same volume of acetonitrile (Sigma-Aldrich, Saint Louis, MO, USA). The mixture was centrifuged again and stained with 1 µL of the supernatant on a polished steel target plate and airdried at room temperature. Then, 1 µL of MALDI matrix (saturated solution of-cyano-4-hydroxycinnamic acid, HCCA, Bruker Daltonik, Bremen, Germany) in 50% acetonitrile and 2.5% trifluoroacetic acid (Sigma-Aldrich, Saint Louis, MO, USA) was applied to each sample. The mass spectrometry results were generated automatically via the Microflex LT MALDI-TOF mass spectrometer (Bruker Daltonik, Bremen, Germany) working in a linearly positive mode in the mass range of 2000–20,000 Da. The device was calibrated using the Bruker bacterial standard. Spectrometric results were processed using MALDI Biotyper 3.0 software (Bruker Daltonik, Bremen, Germany). The following identification criteria were used: a score of 2300 to 3000 indicated highly probable identification at the species level; a score of 2000 to 2299 indicated safe genus identification with probable species identification; and a score of 1700 to 1999 indicated probable identification at the genus level.

### 2.4. The Determination of Kombucha Beverages Polyphenols Profile

Determination of polyphenolic compounds was carried out using the ultra-performance liquid chromatography (UPLC) Waters ACQUITY system (Waters, Milford, MA, USA).

The UPLC system was equipped with a binary pump manager, column manager, sample manager, photodiode array (PDA) detector, and tandem quadrupole mass spectrometer (TQD) with an electrospray ionisation (ESI) source. Separation of polyphenols was performed using a 1.7 µm, 100 mm × 2.1 mm UPLC BEH RP C18 column (Waters, USA). For the anthocyanin investigation, the mobile phase consisted of 2% formic acid in water, *v*/*v* (solvent A) and 2% formic acid in 40% acetonitrile, *v*/*v* (solvent B). However, in the case of other polyphenolic compounds, water (solvent A) and 40% acetonitrile, *v*/*v* (solvent B) were used. The flow rate was kept constant at 0.35 mL/min for a total run time of 8 min. The system was run with the following gradient program: from 0 min 5% B, from 0 to 8 min linear to 100% B, and from 8 to 9.5 min for washing and back to initial conditions. The injection volume of the samples was 5 µL, and the column was supported at 50 °C. The following TQD parameters were used: cone voltage of 30 V, capillary voltage of 3500 V, source and desolvation temperature 120 °C and 350 °C, respectively, and desolvation gas flow rate of 800 L/h. Characterisation of the individual polyphenolic compounds was performed on the basis of the retention time, mass-to-charge ratio, fragment ions, and comparison of data obtained with commercial standards and literature findings. Obtained data were processed in Waters MassLynx v.4.1 software (Waters, Milford, CT, USA).

### 2.5. The Determination of the Kombucha Beverages Mineral Content by Atomic Emission Spectroscopy with Inductively Coupled Plasma (ICP-OES)

The content of mineral elements in the samples was determined by an inductively coupled plasma optical emission spectrometry (ICP-OES) instrument (Schaumburg, IL, USA). Prior to the determination of the elements, the mineralization of the samples was carried out under wet conditions and elevated pressure. Unfiltered samples of kombucha beverages, in the amount of 0.1 mL, were weighed into Teflon containers, and 8 mL of 65% nitric acid was added to it. The samples, thus prepared, were sealed in the vessel. Mineralization of the samples was then performed using a Milestone Ethos Ultrawave-One microwave mineralizer (Milestone SRL, Sorisole, Italy). For each group of nine samples, during the microwave mineralization process, the rotor of the system was supplemented with a blank containing only 8 mL of nitric acid alone. The samples were mineralized for approximately one hour using a temperature rise algorithm as specified for biological samples, without exceeding 200 °C. After cooling, the samples were quantitatively transferred to 50 mL flasks and filled up to the mark with redistilled water. Concentrations of 18 elements (Al, As, Ca, Cd, Cr, Cu, Fe, K, Mg, Mn, Mo, Na, Ni, Pb, Pb, S, Sr, and Zn) were determined by optical emission spectrometry with inductively induced plasma using a Thermo iCAP 6500 Spectrophotometer (Thermo Fisher Scientific Inc., Waltham, MA, USA) (Table 2). The detection threshold obtained for each element was not less than 0.01 mg/kg (with an assumed detection capacity of the measuring instrument at a level higher than 1 ppb). The measurement result for each element was compensated to consider the measurement of the elements in the blank. In each case, a 3-point [13] of 13 calibration curve was used for each element, with optical correction using the internal standard method in the form of yttrium 89Y and ytterbium 173Yb at concentrations of 2 mg/L and 5 mg/L, respectively. The results were expressed in mg/g.

### 2.6. The Determination of pH in Kombucha Beverages

The pH of both the fermented kombucha beverages and the unfermented control was determined by a pH meter S210 (Mettler Toledo SevenCompact™, Columbus, USA). pH determination was of different kombucha varieties on the 0, 1st, 7th, and 14th day of fermentation.

### 2.7. The Determination of Kombucha Alcohol Content

The alcohol content was measured using an alcoholometer (Gomar, Warszawa, Poland). The alcoholometer was immersed in the liquid and the result was read from the scale at room temperature on the 0, 1st, 7th, and 14th day of fermentation.

### 2.8. The Determination of Sugar Content

The total sugar content was measured at room temperature on the 0, 1st, 7th, and 14th day of fermentation with a laboratory refractometer DR6000 (DanLab, Poland) from Brix scale.

### 2.9. Organoleptic Evaluation

The organoleptic evaluation of kombucha tea was carried out by a trained team of 15 evaluators using the 5 points scale method. The sample for analysis consisted of six fermented kombucha beverages variants for 14 days.

The qualitative distinguishing features were as follows.

appearance;colour;aroma;taste;sweetness;acidity.

The following grading scale was used: 1 point—rejecting, 2 points—undesirable, 3 points—satisfactory, 4 points—desirable, and 5 points—very desirable. 

### 2.10. Statistical Analysis

The mean and standard deviation were calculated using Statistica v. 13.1 (StatSoft, Tulsa, OK, USA). The significance of differences between the mean values was verified with the Tukey test (*p* < 0.05).

## 3. Results and Discussion

### 3.1. Determination of the Antibacterial Properties of a Kombucha Beverages

Kombucha tea beverage showed antimicrobial activity against tested yeasts and bacteria species (Table 3). The best antimicrobial activity was determined against Gram-positive bacteria *S. aureus* PCM 502 of 11.0 mm for the white tea + coconut sugar sample, followed by yeast *C. albicans* PCM 1407 of 10.0 mm by the white tea + cane sugar sample (Table 3). Additionally, the green tea + coconut sugar sample showed antimicrobial activity against *C. albicans* PCM 1407 of 10.0 mm. Our results on inhibition of *C. albicans* PCM 1407 growth (10 mm) are comparable with Ref. [14], who identified strong activity (after 18 days of kombucha fermentation) against *C. albicans* (11.00 mm). Fermented tea is recognised as a potential source of antimicrobial and inhibitory substances against various pathogens. Studies show that a kombucha sample containing 7 g/L acetic acid and with a total acid content of 33 g/L has antimicrobial activity against Gram-positive bacteria such as *S. aureus* and *Bacillus cereus* and Gram-negative bacteria such as *E. coli*, *Salmonella choleraesuis serotype typhimurium*, and *Agrobacterium tumefaciens* [6,15]. Kombucha was also found to exert inhibitory effects against Gram-negative bacteria such as *Enterobacter cloacae*, *Pseudomonas aeruginosa*, *Aeromonas hydrophila*, *E. coli*, *Salmonella enteritidis*, *Salmonella typhimurium*, *Yersinia enterocolitica*, *Shigella sonnei*, *Campylobacter jejuni*, and *Helicobacter pylori* and Gram-positive bacteria such as *B. cereus*, *Staphylococcus epidermis*, and *S. aureus* [16,17]. Kombucha prepared from black tea shows the highest antimicrobial activity compared to the beverage prepared from Japanese green tea, mulberry tea, jasmine tea, and oolong tea [18]. Significant antimicrobial activity was also observed in kombucha made from fermentation green tea and black tea for 12 days against Gram-negative rods such as *E. coli*, *P. aeruginosa*, *S. typhimurium* and Gram-positive rods such as *Enterococcus faecalis*, *Micrococcus luteus*, *S. aureus*, and *S. epidermidis* [19]. A recent study showed that black tea kombucha, fermented for 14 days, exhibited maximum antibacterial activity against enterotoxigenic *E. coli*, *Vibrio cholerae*, *Shigella flexneri*, and *S. typhimurium*. Although *V. cholerae* was found to be more susceptible to kombucha, *S. typhimurium* was found to be the most resistant species [2,20]. The inhibitory activity of black tea prepared kombucha has been shown against *C. albicans*, but not against *Zygosaccharomyces bailii* [21,22]. A 21-day fermented beverage of green and black tea inhibited the growth of *C. albicans*, *C. tropicalis*, *Candida parapsilosis*, *C. glabrata*, *Candida dubliniensis*, and *Candida sake*, except *C. krusei* [23]. Acetic acid, which could be major antimicrobial compound, is responsible for the antimicrobial activity of kombucha [17,24]. It is suggested that, in addition to acetic acid or other organic substances, there are other bioactive components such as polyphenols, bacteriocins, proteins, enzymes, etc., which may be responsible for the antimicrobial activity of kombucha [24]. Furthermore, recent studies have shown that the polyphenolic fraction of the extract in ethyl acetate fermented kombucha for 14 days shows significant antimicrobial activity. This finding indicates that the large amount of polyphenols present in the beverage may be responsible for the antimicrobial activity [25,26].

### 3.2. Identification of Kombucha Microflora Using by MALDI-TOF MS Biotyper

Analysis of the composition of the SCOBY microflora using the MALDI TOF MS Biotyper mass spectrometer showed the following microorganisms: *Gluconacetobacter xylinus*, *Acetobacter xylinum*, *Bacterium gluconicum*, *Gluconobacter oxydans*, *Leuconostoc mesenteroides*, *Propionibacterium* spp., *Acetobacter nitrogenifigens*, *Gluconacetobacter kombucha*, *Saccharomyces cerevisiae*, *Candida vini*, *Schizosaccharomyces pombe*, *Pichia membranefaciens*, *Kloeckera apiculate*, *Kluyveromyces marxianus*, and *Pichia kluyveri*
*(*Table 4 and Table 5). Differences in the microflora of the fungus are mainly due to the mixtures of strains used in the production of the SCOBY. The demonstration of differences depends on the climatic conditions, latitude, and the type of microorganisms present and available on the market [27]. 

The tea fungus is not really a ‘fungus’, but a consortium of specially selected bacteria and yeasts that form a uniform culture. They come in various qualitative and quantitative compositions, which undoubtedly depend on the origin of the bacteria and yeast and the type of tea used for fermentation (black, green, or oolong). Our studies have shown that the fermentation time varies between the different types of tea and is faster with black tea infusion under the same fermentation conditions (compared to green tea infusion). Therefore, black tea is the basic raw material in the production of the fermented beverage kombucha. In recent years, attempts have been made to obtain the beverage from other plant raw materials, such as lemon balm, sage, peppermint, fennel, or rosemary. Milk fermented beverages using tea mushroom have also been successfully obtained. Detailed and long term studies on the microbial composition and its differences have shown that the biofilm is composed mainly of acetic bacteria: *A. aceti*, *A. xylinoides*, *A. pasteurianus*, *G. oxydans*, *G. xylinus and yeasts: S. cerevisiae*, *S. ludwigii*, *Z. bailii*, *Brettanomyces bruxellensis*, *Candida famata*, *Pichia membranofacies*, *S. pombe*, or *Torulaspora delbrueckii* [28,29]. Acetic acid bacteria (AAB) are non-forming spore bacilli that occur singly or in chains. They belong to aerobic bacteria. Due to their polar or peritracheal stomata, they are able to move. They are mesophilic bacteria, and their optimal growth temperature is within the range of 25–30 °C. The optimal pH at which these bacteria grow best is between 5 and 6.5. In industry, they are used to produce bacterial cellulose, vinegar, and pharmaceuticals. The biosynthesis of acetic acid is a major component of aerobic metabolism. The substrate in this transformation reaction is ethanol, which is oxidized to acetic acid, where it undergoes further oxidation to water and carbon dioxide in some types of acetic acid bacteria. This process is called complete oxidation of 13 acetic acid [30,31]. The main *Acetobacter* strains carry out superoxidation. When the ethanol resources in the environment are exhausted, peroxidation causes irreversible changes in the metabolism of these bacteria, which is linked to their subsequent inability to oxidise ethanol. Acetic acid bacteria are a group of saprophytic microorganisms but can cause opportunistic infections in immunosuppressed individuals [32,33].

Yeasts are single-celled microorganisms that belong to different taxonomic groups. The most widely used yeasts are yeasts of the genus *Saccharomyces*, especially *S. cerevisiae*. The shape and size of yeast cells are closely related not only to the species and growth conditions of the culture, but also to the age, quantity, and quality of the available food [34]. These factors directly influence the shape of the cell, which can be round, oval, cylindrical, or ellipsoidal. They reproduce vegetatively by growing and producing spores (round or oval), which are contained in bags. They come in two forms: haploid dial and diploid, in which they can grow and develop [13,35]. Haploid yeasts, in the intermediate division process of the cell nucleus, form a new cell that disposes of the same genetic material as the initial cell. This reproduction is called budding. On the other hand, diploid yeast has a double set of homologous chromosomes that come from two different yeast cells that have fused together [36]. They are used to carry out alcoholic fermentation, mainly under anaerobic conditions. Yeast is used in the alcohol, bakery, and dairy industries. From an industrial point of view, there are also yeast species that have harmful effects. Wild yeasts cause product defects, such as bittering and mucilage. The yeast species *S. cerevisiae*, also known as food yeast or noble yeast, is of greatest importance to humans. Yeasts are capable of fermenting and assimilating most sugars and can also use ethanol as a sole carbon source [37,38]. The characteristic feature of this group of yeasts is the ability to perform rapid and dynamic alcoholic fermentation at temperatures up to about 33 °C. The use of mixed yeast cultures in the production of kombucha makes it possible to increase the intensity of the reactions carried out. This is due to the fact that different strains have different characteristics, e.g., resistance to low pH, the ability to ferment quickly, or the ability to ferment sugar to a high degree. Yeast ferments glucose, fructose, sucrose, maltose, lactose, or raffinose [39]. Yeast through invertase (β-fructofuranosidase enzyme) hydrolyzes sucrose to glucose and fructose. The simple sugars produced are further transformed by yeast, which converts them into glycerol and ethanol. Carbon dioxide is also produced during fermentation [31,40]. The resulting ethanol is metabolised to acetic acid by bacteria. Fructose is mainly metabolised to acetic acid. In addition, bacteria from the *Acetobacter* group use the part of glucose that has not been broken down by the yeast to produce organic acids such as glucuronic acid, gluconic acid, and lactic acid as well as cellulose biosynthesis. Cellulose is produced by fermentation with *A. xylinum* and is responsible for the specific appearance of the culture, which forms a 1 cm thick membrane in tea [41].

### 3.3. Identification of Bioactive Compounds Using UPLC-PDA-ESI-MS/MS

Seventeen compounds were determined by the Ultra-Performance Liquid Chromatography method. As a result of chromatographic analysis of ready-made kombucha beverages, 17 bioactive compounds were identified after 14 days of fermentation, i.e., neochlorogenic acid, chlorogenic acid, cryptochlorogenic acid, catechin, gallocatechin 3-*O*-gallate, coumaro-quinic acid, 3-*O*-rithinoside-7-Quercetin O-rhamnoside, Kaemferol 3-*O*-rhamnoside-7-*O*-pentoside, Kaemferol 3-*O*-rhamnoside-7-*O*-pentoside, Quercetin 3-*O*-rutoside-7-*O*-glaucoside, kaemferol 3-*O*-rutoside, Quercetin 3-*O*-rutoside, epicatechin 3-*O*-gallate, Caemferol 3-*O*-rutinoside-7-*O*-rhamnoside, kaemferol 3-*O*-rutinoside, kaemferol 3-*O*-rhamnoside, and kaemferol 3-*O*-rhamnoside (Table 6). This resulted in the development of molecules with antioxidant properties, which are gaining increasing interest as therapeutic agents for diseases related to oxidative stress. Recently, much attention has been given to replacing synthetic antioxidants with natural alternatives such as traditional foods and medicines derived from natural sources containing antioxidants [42]. Many researchers have investigated the antioxidant properties of kombucha. Kombucha made from green tea and black tea has a strong free radical scavenging activity. Kombucha uptake activity has been demonstrated for the DPPH radical, the superoxide radical, and the hydroxyl radical [43]. Recent studies demonstrate the antioxidant capacity of the kombucha beverage prepared from white, green, yellow, and black tea; the highest total phenolic content and the DPPH radical scavenging capacity were detected in yellow tea samples [44]. The activity was also shown to suggest a role in the combat of nitrosative stress. The beverage also showed strong antioxidant activity against lead and chromate [43,45]. In most cases, the antioxidant activity of kombucha was higher than that of unfermented tea. It is hypothesized that some components of low molecular weight are produced, and tea polyphenols are structurally modified by enzymes from the kombucha consortium during fermentation. Kombucha compounds were also found to increase with fermentation time. Thus, the extent of activity depends mainly on the type of tea material, the microbial composition of the kombucha culture, and the fermentation time, which in turn determines the nature of the metabolites. However, prolonged fermentation leads to the accumulation of organic acids, which can be harmful when consumed directly. Furthermore, free radical scavenging properties were found to be reduced when kombucha was stored for 90 days [46]. Therefore, this fermented beverage can be used as a source of antioxidants for many pathophysiological conditions, provided that it is properly prepared and stored.

### 3.4. Determination of the Mineral Content by Atomic Emission Spectroscopy with Inductively Coupled Plasma (ICP-OES)

Fermented kombucha teas contain many elements such as aluminium, calcium, iron, potassium, magnesium, sodium, phosphorus, and sulphur. The analysis was carried out using inductively coupled plasma atomic emission spectroscopy, and the results are presented in Table 7. In all kombucha variants, an increase in the concentration of the tested substances was observed after a 14-day fermentation period, which may indicate the effect of fermentation by kombucha culture on the mineral content of the components. It was observed that all beverages had a high concentration of potassium in relation to other minerals. The kombucha black tea + coconut sugar sample had the highest concentration of this element on the 1st and 14th day of fermentation at 84.38 ppm and 95.97 ppm, respectively. The highest concentration of aluminium was found in the chamber sample green tea + coconut sugar sample after 14 days of fermentation and was 3.068 ppm, and on 1 day it was 1.918 ppm. Calcium concentration was highest on day 14 of fermentation in the kombucha green tea + coconut sugar sample at 11.37 ppm. The highest concentration of magnesium was found in the kombucha black tea + coconut sugar sample after 14 days of fermentation at 7.428 ppm. Sodium concentration was observed on day 1 of fermentation in kombucha black tea + cane sugar sample and was 5.652 ppm; however, after 14 days of fermentation, a decrease in the concentration to 3.12 ppm was observed. The highest concentration of phosphorus was found in the white tea + cane sugar sample after 14 days of fermentation at 10.31 ppm, while the sulphur concentration was the highest in the black tea + coconut sugar sample after 14 days of fermentation and was 11.16 ppm. The exception was iron, and its decrease was observed in most of the samples after the fermentation period, except in the white tea + cane sugar sample. Bauer-Petrovska and Petrushevska-Tozi [47] showed the iron content in fermented tea at the level of 0.353 ppm, which is obvious considering the unstable ionic matrix; they also examined the content of elements such as copper, manganese, nickel, and zinc, and an increase in the tested substances during fermentation was found.

### 3.5. The Analysis of pH, Augar and Alcohol Content in Kombucha Beverages during Fermentation

An important parameter that undergoes change during fermentation is pH. The microorganisms present in SCOBY process the substances included in tea and sugar, producing various metabolites. This is why these parameters change with fermentation time [48]. The present results are similar to the findings of other authors [22]. Chakravorty et al. observed that the initial pH before fermentation was about 5.03 and decreased abruptly to 2.28 after 7 days of fermentation [44]. It has to be remembered that consuming drinks with a very low pH may negatively influence the digestive system [49]. This is why the fermentation time of kombucha is important, as well as the amount of the consumed drink. During the analysis of the pH, it was observed that in all of the tested samples the pH decreased with increasing fermentation time. There were no significant differences in terms of pH between beverages prepared from different types of tea (Table 8). The initial pH for all six samples were quite similar. The highest pH was observed in the black tea + coconut sugar sample for the control day was 5.54. The lowest pH was observed on 14th day for sample green tea + cane sugar was 2.40. The greatest difference between the change in pH was observed in the green tea + cane sugar sample was 3.47.

Refractometric analysis of sugar content showed that all tea types had the highest sucrose concentration before the fermentation process started. In the case of kombucha prepared on the basis of green tea with unrefined cane sugar, the sucrose content decreased with the progress of fermentation, reaching the lowest value on the 14th day of fermentation (5.59 Bx). Sugar content in kombucha also changes in time and depends on fermentation. The initial increase in reducing sugar content can be attributed to the hydrolysis of saccharose into glucose and fructose by yeast. With the progressing fermentation, yeast uses sugar in an oxygen-free way to produce ethanol [50]. In our study, the content of sugar decreased with the time of fermentation. The highest decrease (21.6 %) was observed for the kombucha sample of black tea + coconut sugar on the 14th day of fermentation. Gaggìa et al. checked the content of glucose, fructose, and saccharose in kombucha prepared from black, green, and red tea types on the 7th and 14th day of fermentation. The content of complex carbohydrates, i.e., saccharose, decreased during fermentation, while the content of simple carbohydrates glucose increased. The concentration of fructose increased during fermentation [37]. The highest content of sugars on the 14th day of fermentation was observed in the kombucha sample prepared from white tea and coconut sugar and also in the kombucha green tea + cane sugar sample.

The content of alcohol in all six samples was below 5%, with the highest content of alcohol on 14th day of fermentation at an average content of 4.5%. These results suggest that the kombucha microflora were particularly active. The highest content of alcohol (4.95 %) was observed in the kombucha green tea + coconut sugar sample after 14th days of fermentation. Along with a decrease in the amount of sucrose, the content of ethanol increased with the fermentation time [51].

### 3.6. Organoleptic Evaluation Was Performed after 14-Day Fermentation Period in a Group of 15 People

Subjects rated the appearance, colour, aroma, taste, sweetness, and acidity of six variations of the kombucha beverage (Figure 1). The most desirable appearance was that of kombucha made of green tea and cane sugar, while kombucha made of black tea and coconut sugar had the least desirable appearance due to its slight turbidity and dark colour. The kombucha green tea + cane sugar sample had the most desirable colour, and kombucha of white tea and cane sugar had a less desirable colour; it is largely dependent on the type of tea and sugar used. The aroma that best suited the evaluators was that of kombucha from green tea and coconut sugar, and kombucha from black tea and cane sugar had a less desirable aroma. The kombucha green tea + coconut sugar sample had the best taste, while kombucha sample white tea + cane sugar had the worst taste, the evaluators described the taste as vinegar, slightly tea-like, and comparable to cider. All kombuchas were low in sweetness with high and strongly noticeable acidity [37]. Kombucha from black tea and cane sugar was the sweetest, and the kombucha black tea + coconut sugar sample was the least sweet, while the kombucha white tea + cane sugar sample was the sourest, and the kombucha green tea + coconut sugar sample was also acidic, but to a lesser extent. Vitas et al. [27] prepared yarrow kombucha beverages which had acidic and pleasant taste and odour characteristics for the used extracts. Our results are also similar to the high sensory evaluation of kombucha beverages in the study of Neffe-Skocińska et al. [37].

## 4. Conclusions

In vitro studies confirm the bactericidal and bacteriostatic properties of fermented kombucha beverages, with black and green tea beverages showing the highest antibacterial activity. The bacteria *E. coli* and *Salmonella* sp. were the most sensitive to the effects of kombucha tea beverages. UPLC chromatographic analysis confirmed the presence of 17 bioactive compounds in kombucha beverages that can affect human health. Fermented kombucha teas contained many elements such as aluminium, calcium, iron, potassium, magnesium, sodium, phosphorus, and sulphur. Analysis of the composition of the tea mushroom microflora using the MALDI TOF MS Biotyper mass spectrometer showed the following microorganisms: *Gluconacetobacter xylinus*, *Acetobacter xylinum*, *Bacterium gluconicum*, *Gluconobacter oxydans*, *Leuconostoc mesenteroides*, *Propionibacterium* spp., *Acetobacter nitrogenifigens*, *Gluconacetobacter kombucha*, *Saccharomyces cerevisiae*, *Candida vini*, *Schizosaccharomyces pombe*, *Pichia membranefaciens*, *Kloeckera apiculate*, *Kluyveromyces marxianus*, and *Pichia kluyveri*. The aroma that best suited the evaluators was that of kombucha No. 4 from green tea and coconut sugar, and kombucha No. 1 from black tea and cane sugar had a less desirable aroma.

## Figures and Tables

**Figure 1 foods-11-01523-f001:**
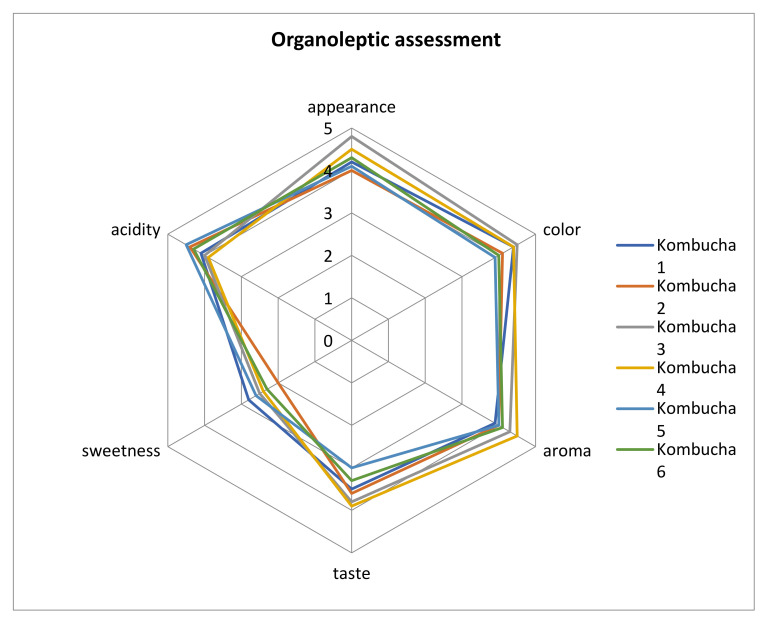
Organoleptic assessment of six variants of kombucha (own elaboration); Samples: 1—Black tea + Cane sugar, 2—Black tea + Coconut sugar, 3—Green tea + Cane sugar, 4—Green tea + Coconut sugar, 5—White tea + Cane sugar, and 6—White tea + Coconut sugar.

**Table 1 foods-11-01523-t001:** The composition of the prepared kombucha beverages.

No. Trials	The Composition of the Beverage
1	Black tea + Cane sugar
2	Black tea + Coconut sugar
3	Green tea + Cane sugar
4	Green tea + Coconut sugar
5	White tea + Cane sugar
6	White tea + Coconut sugar

**Table 2 foods-11-01523-t002:** Measurement parameters and recovery rates of individual elements.

Element	Measurement Linse (nm)	Recovery by CRM (%)	Recovery by (%)
Al	167.079	98	100
Ca	317.933	101	99
Cu	324.754	98	99
K	766.490	102	98
Mg	279.533	102	101
P	177.495	101	99
S	180.731	97	100
Zn	213.856	99	97

The detection limit for each element was determined to be no less than 1 µg/L.

**Table 3 foods-11-01523-t003:** Inhibition zone of growth inhibition for different species of bacteria and yeast (mm).

	Gram-Negative Bacteria		Gram-Positive Bacteria		Yeasts			
Type of Beverage	ST	EC	LM	SA	CG	CA	CK	CT
Black tea + cane sugar	1.3 *^a^* ± 0.6	1.0 *^a^* ± 0.0	3.0 *^c^* ± 0.0	1.0 *^a^* ± 0.0	2.0 *^b^* ± 0.0	2.0 *^b^* ± 0.0	6.3 *^d^* ± 0.6	9.7 *^e^* ± 0.6
Black tea + coconut sugar	3.6 *^c^* ± 0.6	3.0 *^c^* ± 0.0	3.0 *^c^* ± 0.0	1.0 *^a^* ± 0.0	2.0 *^b^* ± 0.0	3.0 *^c^* ± 0.0	7.5 *^d^* ± 0.6	8.4 *^d^* ± 0.6
Green tea + cane sugar	9.0 *^c^*± 0.2	7.0 *^b^*± 0.5	8.0 *^c^* ± 0.1	5.3 *^a^* ± 0.6	7.3 *^b^* ± 0.6	6.7 *^b^* ± 0.0	9.7 *^d^* ± 0.2	6.0 *^a^* ± 0.0
Green tea + coconut sugar	4.3 *^a^* ± 0.6	6.1 *^b^* ± 0.1	6.0 *^b^* ± 0.5	7.7 *^c^* ± 0.6	7.8 *^c^* ± 1.0	8.3 *^c^* ± 0.7	4.7 *^a^* ± 0.6	7.3 *^c^* ± 0.3
White tea + cane sugar	5.3 *^a^*± 0.6	7.7 *^b^* ± 0.5	8.0 *^b^* ± 1.0	10.3 *^c^* ± 0.6	8.0 *^b^* ± 1.0	10.0 *^c^* ± 0.3	5.3 *^a^* ± 0.6	7.7 *^b^* ± 0.5
White tea + coconut sugar	5.2 *^a^* ± 1.0	7.1 *^b^* ± 0.0	7.5 *^b^* ± 0.1	11.0 *^d^* ± 0.5	9.0 *^c^* ± 0.4	9.0 *^c^* ± 0.1	5.4 *^a^*± 1.0	6.5 *^a^* ± 0.3

Note: Means ± standard deviation. Variants not sharing the same letter(s) are statistically significant at *p* < 0.05 (Tukey’s HSD). *S. typhi* PCM 2548—ST, *E. coli* 2209—EC, *L. monocytogenes* PCM 2606—LM, *S. aureus* PCM 502—SA, *C. glabrata* PCM 2703-FY—CG, *C. albicans* PCM 1407—CA, *C. krusei* PCM 2706-FY —CK, and *C. tropicalis* PCM 2703-FY—CT. ATB—positive control (Cefoxitin for G^−^, Gentamicin for G^+^, Fluconazole for yeast. ). *^a–e^*—mean values denoted in rows by different letters differ statistically significantly at *p* ≤ 0.05.

**Table 4 foods-11-01523-t004:** Identification of bacteria using by MALDI-TOF MS Biotyper.

Microorganism	Gram Negative	Gram Positive	Black Tea + Cane Sugar	Black Tea + Coconut Sugar	Green Tea + Cane Sugar	Green Tea + Coconut Sugar	White Tea + Cane Sugar	White Tea + Coconut Sugar
*Gluconacetobacter xylinus*	+	−	−	+	−	+	−	+
*Acetobacter xylinum*	+	−	+	−	+	−	+	−
*Bacterium gluconicum*	+	−	+	−	+	−	+	−
*Gluconobacter oxydans*	+	−	+	+	+	+	−	−
*Leuconostoc mesenteroides*	−	+	+	+	+	+	−	−
*Propionibacterium* spp.	−	+	+	+	−	−	+	+
*Acetobacter nitrogenifigens*	+	−	−	−	+	+	+	+
*Gluconacetobacter kombucha*	+	−	+	+	+	+	+	+

**Table 5 foods-11-01523-t005:** Identification of yeasts using by MALDI-TOF MS Biotyper.

Microorganism	Black Tea + Cane Sugar	Black Tea + Coconut Sugar	Green Tea + Cane Sugar	Green Tea + Coconut Sugar	White Tea + Cane Sugar	White Tea + Coconut Sugar
*Saccharomyces cerevisiae*	+	+	+	+	+	+
*Candida vini*	+	−	+	−	+	−
*Schizosaccharomyces pombe*	−	+	−	+	−	+
*Pichia membranefaciens*	−	+	−	+	−	+
*Kloeckera apiculate*	+	−	+	−	+	−
*Kluyveromyces marxianus*	+	+	−	+	−	+
*Pichia kluyveri*	+	−	−	−	+	−

**Table 6 foods-11-01523-t006:** Bioactive compounds identified during UPLC analysis after 14 days of fermentation.

No.	Compound	Rt * (min)	UV–Vis * λ_max_	[M−H]^−^ *m*/*z* *	MS/MS *
1	Neochlorogenic acid	2.79	299, 327	353	191
2	Chlorogenic acid	3.55	299, 327	353	191
3	Cryptochlorogenic acid	3.72	299, 327	353	191
4	Catechin	4.24	274	289	151
5	Gallocatechin 3-*O*-gallate	4.33	274	457	305
6	Coumaroyl quinic acid	4.44	299, 311	337	191
7	Quercetin 3-*O*-rutinoside-7-*O*-rhamnoside	4.74	255, 350	755	609, 301
8	Kaempferol 3-*O*-rhamnoside-7-*O*-pentoside	4.83	264, 355	563	447, 285
9	Kaempferol 3-*O*-rhamnoside-7-*O*-pentoside	4.89	264, 355	563	447, 285
10	Quercetin 3-*O*-rutinoside-7-*O*-galactoside	5.2	255, 350	771	609, 301
11	Kaempferol 3-*O*-rutinoside	5.35	264, 355	593	285
12	Quercetin 3-*O*-rutinoside	4.46	255, 355	609	301
13	Epicatechin 3-*O*-gallate	5.56	278	441	305
14	Procyanidin A1	5.88	274	577	289
15	Kaempferol 3-*O*-rutinoside-7-*O*-rhamnoside	5.97	264, 350	739	593, 285
16	Kaempferol 3-*O*-glucoside-rhamnoside	6.08	264, 355	593	285
17	Kaempferol 3-*O*-rhamnoside	6.39	265, 352	447	285

Note: * Rt—retention time, UV–Vis spectrum, [M−H]^−^—negative ion mode, MS—mass spectrometry.

**Table 7 foods-11-01523-t007:** Comparison of the elemental composition of different variances of kombucha on the 1st and 14th day of fermentation.

Sample	Day	Al	Ca	Fe	K	Mg	Na	P	S
Black tea + Cane sugar	1	1.52 *^e^*	3.119 *^cd^*	0.1281 *^g^*	79.01 *^d^*	5.323 *^h^*	5.652 *^h^*	6.637 *^d^*	5.362 *^c^*
		±0.033	±0.027	±0.012	±0.68	±0.053	±0.064	±0.08	±0.026
	14	2.102 *^f^*	8.087 *^e^*	0 *^d^*	69.6 *^ef^*	6.066 *^g^*	3.12 *^g^*	6.448 *^d^*	5.406 *^cd^*
		±0.217	±0.047	±0.072	±0.24	±0.4	±0.031	±0.061	±0.003
Black tea + Coconut sugar	1	1.678 *^bc^*	9.268 *^f^*	0.1032 *^i^*	84.38 *^g^*	6.499 *^e^*	1.859 *^e^*	6.634 *^d^*	9.567 *^fg^*
		±0.101	±0.062	±0.0066	±0.46	±0.036	±0.013	±0.024	±0.074
	14	2.51 *^d^*	10.75 *^i^*	0.0692 *^i^*	95.9 *^j^*	7.428 *^f^*	2.203 *^f^*	8.859 *^e^*	11.16 *^i^*
		±0.105	±0.16	±0.0269	±0.0015	±0.1104	±0.024	±0.051	±0.02
Green tea + Cane sugar	1	1.559 *^b^*	2.119 *^a^*	0.0466 *^a^*	50.28 *^a^*	2.65 *^a^*	0.658 *^a^*	4.201 *^a^*	4.012 *^a^*
		±0.128	±0.02	±0.0002	±0.51	±0.031	±0.011	±0.054	±0.013
	14	2.617 *^d^*	2.525 *^b^*	0.0334 *^b^*	56.05 *^b^*	3.002 *^bcd^*	1.014 *^bcd^*	5.332 *^b^*	4.531 *^b^*
		±0.167	±0.056	±0.0337	±0.84	±0.051	±0.018	±0.037	±0.017
Green tea + Coconut sugar	1	1.918 *^c^*	10.21 *^g^*	0.1393 *^c^*	68.24 *^c^*	4.652 *^ab^*	0.8265 *^ab^*	5.323 *^b^*	9.302 *^f^*
		±0.14	±0.08	±0.0308	±0.24	±0.069	±0.0122	±0.062	±0.029
	14	3.068 *^e^*	11.37 *^j^*	0.1067 *^e^*	75.49 *^d^*	5.143 *^abc^*	0.8538 *^abc^*	6.111 *^c^*	9.997 *^h^*
		±0.236	±0.13	±0.0446	±0.57	±0.016	±0.0158	±0.39	±0.051
White tea + Cane sugar	1	0.874 *^a^*	2.905 *^c^*	0.063 *^fg^*	78.93 *^e^*	5.806 *^d^*	1.225 *^d^*	8.865 *^e^*	5.697 *^d^*
		±0.145	±0.11	±0.0861	±0.78	±0.04	±0.009	±0.028	±0.028
	14	1.572 *^b^*	3.201 *^d^*	0.087 *^h^*	83.85 *^f^*	6.156 *^cd^*	1.148 *^cd^*	10.31 *^g^*	6.108 *^e^*
		±0.129	±0.042	±0.0301	±1.01	±0.103	±0.016	±0.07	±0.014
White tea + Coconut sugar	1	1.04 *^a^*	10.57 *^hi^*	0.1597 *^f^*	78.67 *^h^*	6.647 *^d^*	1.223 *^d^*	9.042 *^e^*	9.686 *^g^*
		±0.117	±0.041	±0.0523	±0.55	±0.064	±0.026	±0.033	±0.07
	14	1.456 *^b^*	10.41 *^gh^*	0.0726 *^h^*	83.64 *^i^*	6.943 *^ef^*	2.109 *^ef^*	9.641 *^f^*	10.09 *^h^*
		±0.16	±0.052	±0.0637	±0.16	±0.039	±0.004	±0.062	±0.02

Note: Means ± standard deviation. Variants not sharing the same letter(s) are statistically significant at *p* < 0.05 (Tukey’s HSD). *^a–i^*—mean values denoted in rows by different letters differ statistically significantly at *p* ≤ 0.05.

**Table 8 foods-11-01523-t008:** Comparison of alcohol, pH, and sugar content of different kombucha varieties on the 0, 1st, 7th, and 14th day of fermentation.

Sample	Day	Alcohol (%)	pH	Saccharose [Brix-g/100 mL]
Black tea + Cane sugar	0	0.0 ± 0.00	5.87 ± 0.02	7.04 ± 0.09
	1	0.3 ± 0.00	3.35 ± 0.01	6.95 ± 0.35
	7	3.5 ± 0.50	2.60 ± 0.02	6.78 ± 0.00
	14	4.85 ± 0.50	2.44 ± 0.02	5.64 ± 0.00
Black tea + Coconut sugar	0	0.0 ± 0.00	5.54 ± 0.02	7.06 ± 0.02
	1	0.4 ± 0.00	3.51 ± 0.04	6.65 ± 0.25
	7	3.25 ± 0.00	2.63 ± 0.03	6.22 ± 0.00
	14	4.00 ± 0.00	2.58 ± 0.02	5.59 ± 0.00
Green tea + Cane sugar	0	0.0 ± 0.00	5.87 ± 0.03	7.13 ± 0.09
	1	0.3 ± 0.50	3.54 ± 0.04	6.73 ± 0.35
	7	3.4 ± 0.00	2.61 ± 0.03	6.35 ± 0.00
	14	4.00 ± 0.00	2.40 ± 0.02	5.72 ± 0.00
Green tea + Coconut sugar	0	0.0 ± 0.00	5.62 ± 0.04	7.11 ± 0.09
	1	0.2 ± 0.00	3.75 ± 0.03	6.79 ± 0.35
	7	3.50 ± 0.50	2.60 ± 0.02	6.46 ± 0.00
	14	4.95 ± 0.50	2.56 ± 0.02	5.70 ± 0.00
White tea + Cane sugar	0	0.0 ± 0.00	5.64 ± 0.04	7.15 ± 0.09
	1	0.4 ± 0.00	3.72 ± 0.02	6.59 ± 0.35
	7	3.0 ± 0.00	2.81 ± 0.01	6.47 ± 0.00
	14	4.50 ± 0.00	2.72 ± 0.02	5.64 ± 0.00
White tea + Coconut sugar	0	0.0 ± 0.00	5.70 ± 0.09	7.17 ± 0.09
	1	0.2 ± 0.50	3.25 ± 0.35	6.63 ± 0.35
	7	3.25 ± 0.50	2.74 ± 0.09	6.29 ± 0.00
	14	4.75 ± 0.00	2.65 ± 0.35	5.84 ± 0.00

## Data Availability

The data are contained within the article.

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
