# Peer review of "Microbiological and Physicochemical Composition of Various Types of Homemade Kombucha Beverages Using Alternative Kinds of Sugars"

_foods, 2022, doi:10.3390/foods11101523_

Round 1

Reviewer 1 Report

The text need revisions concerning the english and format, additionally major revisions concerning presentation of methods used, presentation and discussion of the results are necessary.

Author Response

Dear Reviewer, 

thank you for your time, comments and sugestives. I correct and added your opinion to text (red color). 

I send you draft. 

best Regards, 

MK 

Reviewer 2 Report

  • The differential of the study was to analyze kombuchas prepared from two different types of sugars, however, the authors did not discuss their results on this aspect. The differences between the kombuchas produced by the two types of sugars were only superficially discussed in the topic “Organoleptic evaluation was performed after 14-day fermentation period in a group of 15 416 people”.
  • The abstract presents incoherent information, for example, that one of the objectives was to analyze the phenolic compounds formed during the fermentation process. In fact, the phenolic compounds present in kombucha come from tea (plant leaves), changes in the structure and content of these compounds may occur during the fermentation process, however, they are not produced during the fermentation process.
  • The author reports that kombucha is recommended for various diseases including AIDS. This information is completely wrong and it is a very serious error. The word “aids” was used by the referenced authors with another meaning (“help”) and not as the disease “Acquired Immunodeficiency Syndrome”, which the acronym is AIDS.
  • Line 49 and 50 the author reports that kombucha has up to 2% alcohol, however, in the results presented by the authors, kombucha had an alcohol content well above 2% and none of this was discussed in the manuscript.
  • The author did not include a quote in lines 52 and 75.
  • Line 81 – in the acronym “SCOOBY” there is an extra letter "o".
  • In the topic “Preparation of the kombucha beverages,” the authors need to elaborate further on the next steps after the addition of the SCOBY culture, at what temperature was the kombucha incubated? In addition, the authors do not cite any bibliographic references. The kombuchas were prepared based on some references, so this must be cited and referenced in the article.
  • In the topic “Antibacterial activity of the kombucha beverages” no reference method was cited for this analysis. The authors cite tests with yeast (Candida), so the topic title is inappropriate. In addition, the antimicrobial analysis must follow a reference (international standard) and the same culture medium must not be used for microorganisms with different characteristics (bacteria and yeasts). This topic includes repeated phrases.
  • The results were not well discussed, the figure was not cited in the text, and tables must be self-explanatory and well-formatted, in addition to other writing problems that compromise the work. Authors should do a detailed and thorough review of the manuscript before resubmitting it to a high-impact journal.

Author Response

Dear Reviewer, 

thank you for your time, comments and sugestives. I correct and added your opinions to text (red color). 

I send you draft. 

best Regards, 

MK 

Reviewer 3 Report

Thank you very much for your interesting research. Some points must be carefully revised:

  1. Lines 42-44. As far as I know, the terms “tea fungus”, “tea mushroom” or “kombucha mushroom” are referred to the SCOBY i.e. the culture. However, the “kombucha” beverage would be the result of the fermentative activity of the SCOBY. In my opinion, these sentences must be rephrased for better understanding. Useful reference: https://doi.org/10.1111/1541-4337.12073
  2. Line 49. Also vitamin C.
  3. Lines 53-55. It is crucial to include an additional statement for this introductive part: although the effects observed in vitro are promising, more in vivo tests are required and clinical trials must be carried out to validate these results. You can use this reference: https://doi.org/10.1016/j.tifs.2020.09.025
  4. Lines 76-79. From my point of view, this paragraph must be rephrased. Saying that kombucha beverage is recommended (for all these relevant health disorders) without clinical evidence supported by human trials sounds speculative.
  5. MATERIAL AND METHODS. Lines 86-87. Please, indicate the scientific name of the used teas (Camellia sinensis or other).
  6. MATERIALS AND METHODS. Line 92. Please, indicate if the fermentation was carried out at room temperature, as well as the number of days and the light or darkness conditions.
  7. RESULTS & DISCUSSION. Line 227. The cited work (reference number 15) used probiotic-, prebiotic- and antibiotic-supplemented-diets in chickens. However, I could not find the presence of fermented tea in the study, so it must be rephrased for a better understanding.
  8. RESULTS & DISCUSSION. Lines 262-263. The term “mushroom” is used throughout the text and it can be confusing for readers. It must be clarified that it is not really a mushroom. In this context, “mushroom” is just an adapted name for the SCOBY. This clarification is included in lines 262-263 but, in my opinion, it should be explained in the introduction section to avoid confusion.
  9. RESULTS & DISCUSSION. Could you please add the statistical analysis for the quantitative results compilated in Table 7 & Table 8? Letters indicating significant differences (as you did for Table 3) will be useful and explanatory.

Author Response

(The authors gave the same response as above.)

Reviewer 4 Report

These are my comment on the paper entitled "Microbiological and physicochemical composition of various types of homemade kombucha beverages using alternative kinds of sugars".

  • work is scientific-relevant, used methods are appropriate; the obtained results are very promising;
  • Please, in the Materials and methods emphasize the origin of kombucha, inoculum preparation, as well as incubation conditions. Emphasize the manner and level of inoculation of selected teas.
  • 2.2. section - please, add references and explain in detail used methodology; line 105 - did you use 105 CFU/mL or 5 log CFU/mL? Correct it. Also, this is the firt place where you list test microorganisms, so you need to use full name for them.
  • The only problem I have with the obtained results is antimicrobial testing. The obtained inhibition zones are minimal, so the antimicrobial effect of the obtained kombucha is not at an enviable level. So, I required that the Authors better explain the obtained results, and also compare them with scientific-relevant papers.
  • Table 3 - indicate what are letters a,b,c
  • in vitro must be italic throughout the whole manuscript
  • table 6 - the Authors used abbreviations for parameters, please add the whole names, or indicate in footnotes what are abbreviations Rt, UV...
  • table 7 - what represents the numbers in brackets?

Author Response

(The authors gave the same response as above.)

Round 2

Reviewer 1 Report

The authors presented a review of the manuscript, both in English and of the information pointed in the previous review.

Author Response

Dear Reviewer, 

Thank you for your opinion and work, which was helped to corrected the manuscript. 

Best Regards, 

Maciej Kluz

Reviewer 2 Report

I did not receive a response letter from the authors, I received a new version of the manuscript in which most of my comments were not considered.

Author Response

Dear Reviewer,

Thank you for your opinion and suggestion. I try to corrected all of them.

Best Regards,

MK

  1. Lines 42-44. As far as I know, the terms “tea fungus”, “tea mushroom” or “kombucha mushroom” are referred to the SCOBY i.e. the culture. However, the “kombucha” beverage would be the result of the fermentative activity of the SCOBY. In my opinion, these sentences must be rephrased for better understanding. Useful reference: https://doi.org/10.1111/1541-4337.12073
  2. Thank you for your suggestion. I changed in text for SCOBY and used reference number 1.
  3. Line 49. Also vitamin C.
  4. Corrected.
  5. Lines 53-55. It is crucial to include an additional statement for this introductive part: although the effects observed in vitro are promising, more in vivo tests are required and clinical trials must be carried out to validate these results. You can use this reference: https://doi.org/10.1016/j.tifs.2020.09.025

Corrected and added reference – number 3.

  1. Lines 76-79. From my point of view, this paragraph must be rephrased. Saying that kombucha beverage is recommended (for all these relevant health disorders) without clinical evidence supported by human trials sounds speculative.
  2. Corrected
  3. MATERIAL AND METHODS. Lines 86-87. Please, indicate the scientific name of the used teas (Camellia sinensis or other).
  4. Corrected
  5. MATERIALS AND METHODS. Line 92. Please, indicate if the fermentation was carried out at room temperature, as well as the number of days and the light or darkness conditions.
  6. Corredcted
  7. RESULTS & DISCUSSION. Line 227. The cited work (reference number 15) used probiotic-, prebiotic- and antibiotic-supplemented-diets in chickens. However, I could not find the presence of fermented tea in the study, so it must be rephrased for a better understanding.
  8. Corrected
  9. RESULTS & DISCUSSION. Lines 262-263. The term “mushroom” is used throughout the text and it can be confusing for readers. It must be clarified that it is not really a mushroom. In this context, “mushroom” is just an adapted name for the SCOBY. This clarification is included in lines 262-263 but, in my opinion, it should be explained in the introduction section to avoid confusion.
  10. Corrected
  11. RESULTS & DISCUSSION. Could you please add the statistical analysis for the quantitative results compilated in Table 7 & Table 8? Letters indicating significant differences (as you did for Table 3) will be useful and explanatory.
  12. Corrected.

Reviewer 4 Report

After reviewing the revised version, I can suggest the acceptance of the paper.

Author Response

(The authors gave the same response as above.)
